# Blue and Red LED Lights Differently Affect Growth Responses and Biochemical Parameters in Lentil (*Lens culinaris*)

**DOI:** 10.3390/biology13010012

**Published:** 2023-12-24

**Authors:** Benedetta Bottiglione, Alessandra Villani, Linda Mastropasqua, Silvana De Leonardis, Costantino Paciolla

**Affiliations:** 1Department of Biosciences, Biotechnology and Environment, University of Bari Aldo Moro, Via E. Orabona 4, 70125 Bari, Italy; benedetta.bottiglione@uniba.it (B.B.); linda.mastropasqua@uniba.it (L.M.); silvana.deleonardis@uniba.it (S.D.L.); 2Institute of Sciences of Food Production, National Research Council of Italy, Via G. Amendola, 122/O, 70126 Bari, Italy

**Keywords:** *Lens culinaris* Medik., monochromatic LED light, light intensity, ascorbic acid, photosynthetic pigments, phenols, antioxidant enzymes, biometric parameters

## Abstract

**Simple Summary:**

Light-emitting diodes (LEDs) offer several advantages compared to conventional light sources. These systems are considered environmentally friendly tools to improve plant product quality. Controlling the light environment using this technology has shown positive effects, such as the potential to enhance plant photosynthesis and productivity, modulate the synthesis of bioactive compounds, extend the shelf life of fruits and vegetables, and increase resistance to biotic and abiotic stress. Among others, the possibility of using a narrow-band spectrum allows us to investigate the effects of single monochromatic light. In this study, we evaluated changes in some of the biometrical and biochemical parameters in lentil seedlings (*Lens culinaris* Medik.) grown under two wavelengths and three light intensities provided by LED lamps. Red light proved to have a strong effect on seedling elongation. In contrast, blue light affected the content of some bioactive compounds and enzymes. Research in plant–LED light interaction can provide insights for applications in agricultural conditions or indoor cultivations and contribute to understanding how light regulates plant growth and development.

**Abstract:**

Light-emitting diodes are an attractive tool for improving the yield and quality of plant products. This study investigated the effect of different light intensity and spectral composition on the growth, bioactive compound content, and antioxidant metabolism of lentil (*Lens culinaris* Medik.) seedlings after 3 and 5 days of LED treatment. Two monochromatic light quality × three light intensity treatments were tested: red light (RL) and blue light (BL) at photosynthetic photon flux density (PPFD) of 100, 300, and 500 μmol m^−2^ s^−1^. Both light quality and intensity did not affect germination. At both harvest times, the length of seedling growth under BL appeared to decrease, while RL stimulated the growth with an average increase of 26.7% and 62% compared to BL and seedlings grown in the darkness (D). A significant blue light effect was detected on ascorbate reduced form, with an average increase of 35% and 50% compared to RL-grown plantlets in the two days of harvesting, respectively. The content of chlorophyll and carotenoids largely varied according to the wavelength and intensity applied and the age of the seedlings. Lipid peroxidation increased with increasing light intensity in both treatments, and a strong H_2_O_2_ formation occurred in BL. These results suggest that red light can promote the elongation of lentil seedlings, while blue light enhances the bioactive compounds and the antioxidant responses.

## 1. Introduction

The interest in fresh sprout products has been increasing over the last two decades among health-conscious consumers due to their outstanding source of nutrients and content of beneficial secondary metabolites such as glucosinolates, phenols, and vitamins [1]. Sprouts are harvested shortly after germination when various physiological and biochemical processes occur, including the transformation of macromolecules into more digestible components, such as oligosaccharides, free amino acids, and organic acids, and the increase in bioactive compounds, antioxidant capacity, and mineral bioavailability [1,2]. Several studies showed that the growth of hypocotyls, the phytochemical compound content, and the antioxidant capacity of sprouts are generally affected by environmental and genetic factors, among which light can play a crucial role [2]. Although sprouting usually occurs in the darkness, an increasing number of studies have revealed that artificial light sources could enhance the nutritional value of these foodstuffs [3,4,5,6,7].

Light can determine different responses in plants, modulating photosynthesis, seed germination, and plant development during the ontogenetic cycle [8,9]. These responses are mediated by specific reception and signalling systems, including photoreceptors and pigments that can capture different light wavelengths and irradiance levels. The photoreceptors are divided into phytochromes, which are known as primary red/far-red light receptors; cryptochromes, phototropin, and the ZTL/FKF1/LKP2 complex, which absorb blue and UV-A wavelengths; UV-B resistance locus 8 is sensitive to UV-B wavelengths [8,10,11]. The chlorophyll and carotenoid pigments are involved in the photosynthetic processes and photoprotection, respectively [12,13].

In the last two decades, several studies focused on the interaction between light and metabolic pathways in plants. Due to such implications, different artificial light sources have commonly been used [11,14].

Nowadays, LED technology has gained significant popularity, and its application has increased due to the many advantages compared to other conventional light sources, such as fluorescent, high-pressure mercury, high-pressure sodium, and metal-halide lamps [11]. LEDs show unique and flexible spectral use in terms of wavelengths and intensities, as well as high-energy efficiency, lower operational cost, low radiant heat, and longer lifespan [14]. Moreover, the narrow-waveband spectra emission provided by LED, and the opportunity to regulate light intensity according to the horticulture S640 metric standards published by the American Society of Agricultural and Biological Engineers [ASABE, 2017] may allow for a deeper evaluation of the effect of light on plant growth and development and better definition of species-specific light recipes.

Currently, blue and red lights are widely used in horticulture to investigate plant responses to light, highlighting species-specific responses [15,16,17,18,19].

Molecular and physiological studies showed that red light enhances photosynthesis and vegetative growth, even though prolonged monochromatic red light may cause the so-called “red light syndrome” characterized by leaf curling and decreased photosynthetic capacity, leaf thickness, and leaf pigmentation [20]. On the other hand, monochromatic blue light may increase the hypocotyl length and chlorophyll content of several plant species, including buckwheat [21], eggplant [16], cherry tomato [17], and different species of the genus *Brassica* [18]. Furthermore, intensity could play a pivotal role in the regulation of the photosynthetic process and plant development. Recently, Johnson et al. [22] have demonstrated that blue light promoted the hypocotyl elongation of arugula at various PPFD levels (from 20 up to 650 μmol m^−2^ s^−1^), even though promotion decreased with increased PPFD levels. On the contrary, the promotion effect of pure blue light on mustard was only detected at higher PPFD levels.

Establishing whether light intensity may have a more relevant role than light quality on plant growth and development, and how these effects may vary among species and even cultivars, is still challenging. Moreover, few studies so far have investigated the effect of light quantity and quality on biochemical composition and antioxidant activities [3,23,24].

Based on the above information, the aim of this study was to explore the effect of pure blue and red light at different intensities on the growth, the content of some bioactive compounds, and the antioxidant capacities of lentil seedlings.

Pro-health products, such as legumes, are commonly used for sprouting and microgreen production due to their high nutritional value and abundance of minerals, bioactive compounds, and antioxidants (e.g., phenolics and vitamins) [25]. Biometric (fresh and dry weight, root and shoot length) and biochemical parameters (phenols, chlorophylls, carotenoids, ascorbate, antioxidant enzymes, and oxidative stress markers) have been investigated.

## 2. Materials and Methods

### 2.1. Plant Material and Growth Conditions

Lentil (*Lens culinaris* Medik.) seeds were selected as experimental material and were purchased from the commercial source Alce Nero S.p.A. (Bologna, Italy). Seeds (30 g) were surface sterilized in 0.5% (*v*/*v*) sodium hypochlorite solution for 5 min, rinsed in distilled water (dH_2_O) until wash water reached pH 7, and soaked in dH_2_O for three hours via gentle mechanical agitation allowing for gas exchange. All actions were performed far from direct light sources. The seeds were evenly spread on the grid plate of seed-sprouting trays (32.5 × 24.5 × 4.5 cm, length × width × height) filled with only distilled water. Trays were then transferred to a walk-in growth chamber for germination and growth. The chamber was equipped with two light-emitting diode (LED) bar lamps for light treatments. Lentil seeds were germinated and grown in darkness (D) and under continuous red (RL) and blue (BL) LED light up to five days after sowing. During the experiment, the day and night temperature was 24 ± 1 °C and the relative humidity was 55–60%. Seedlings were harvested at 3 and 5 days after sowing (DAS). Ten seedlings per treatment were randomly collected and used for biometrical measurements. Batches of about 0.5 g seedlings were collected, put in a foil paper bag, immediately frozen in liquid nitrogen, and stored at −80 °C for biochemical analysis.

### 2.2. Light Treatments

Lentil seeds were germinated and grown under continuous light provided by two monochromatic LED lights at three photosynthetic photon flux densities (PPFD) of 100, 300, and 500 μmol m^−2^ s^−1^ in continuous darkness. The light wavelength peaks were 470 nm (Blue LED) and 630 nm (Red LED) and were provided by LED AE80 lamps (Ambra Elettronica S.r.l., Vicenza, Italy). The PPFD for each light source was measured with a photo radiometer, Model HD2302.0 (Delta OHM S.r.l., Padova, Italy).

### 2.3. Biometric Measurements

Ten seedlings per treatment were collected and used to determine the fresh weight (FW) and the dry weight (DW) using an electronic analytical balance. After FW determination, samples were dried at 75 ± 2 °C in an oven until a constant weight was reached (~48 h) to obtain the DW. Dry matter content (DM) was obtained by calculating the ratio between DW and FW expressed as a percentage (%). For total, aerial part and root length determination, ten seedlings randomly sampled were analyzed using ImageJ software 1.53v.

### 2.4. Chlorophyll and Carotenoid Contents

The chlorophyll and carotenoid contents were determined according to Lichtenthaler [12] with some modifications. In detail, 0.5 g of each sample was ground with 8 mL of absolute acetone and centrifuged at 20,000× *g* for 15 min. The absorbance of the supernatant was spectrophotometrically measured at 645 nm for chlorophyll *a*, 662 nm for chlorophyll *b*, and 470 nm for carotenoids. Chlorophyll (*a*, *b*, and *a* + *b*) and carotenoid contents were calculated using the following equations:Chl *a* (µg/mL) = 11.24 OD_662_ − 2.04 OD_645_
Chl *b* (µg/mL) = 20.13 OD_645_ − 4.19 OD_662_
Chls (*a* + *b*) (µg/mL) = 7.05 OD_662_ + 18.09 OD_645_
Car (µg/mL) = (1000 OD_470_ − 1.90 Chl *a* − 63.14 Chl *b*)/214

The calculated value was expressed as µg/g DW.

### 2.5. Total Phenols

Total phenols were determined using the Folin–Ciocalteu method, as previously described by Villani et al. [26]. Phenols content was measured by homogenizing 0.5 g of plant material with 80% ethanol solution in a 1:8 weight/volume ratio. The homogenate was centrifuged at 7000× *g* for 10 min, and 0.05 mL of supernatant was mixed with 0.95 mL of _d_H_2_O and 0.05 mL of a 1:1 water diluted Folin–Ciocalteu reagent. After 3 min, 100 mL of a 0.1 M NaOH solution containing 20% (*w*/*v*) Na_2_CO_3_ was added, and the mixture was incubated at 25 °C for 90 min in darkness. The absorbance was spectrophotometrically measured at 760 nm. Total phenols were expressed as mg of gallic acid equivalent (GAE) on dry weight (g DW).

### 2.6. Ascorbate Pool

For ascorbate (AsA) and dehydroascorbate (DHA) content determination, 0.5 g of plant material was homogenized with four volumes of cold 5% (*w*/*v*) metaphosphoric acid and centrifuged at 20,000× *g* for 15 min. The resulting supernatant was analyzed to assay AsA, DHA, and total AsA (AsA + DHA) according to Paciolla et al. [27].

### 2.7. Hydrogen Peroxide and Lipid Peroxidation Determination

Hydrogen peroxide (H_2_O_2_) content was evaluated according to the method described by Singh et al. [28] with some modifications. The samples (0.5 g) were homogenized with four volumes of 0.1% (*w*/*v*) trichloroacetic acid (TCA) and centrifuged at 12,000× *g* for 15 min at 4 °C. A total of 0.5 mL of supernatant was added to 0.5 mL of 10 mM potassium–phosphate buffer (pH 7.0) and 1 mL of 1 M potassium iodide and incubated for 1 h in the dark at 25 °C. The absorbance was measured spectrophotometrically at 390 nm.

Lipid peroxidation was assessed in terms of malondialdehyde (MDA) content by measuring the TBA–MDA complex, as described by Zhang and Kirkham [29]. Plant material (0.5 g) was ground with four volumes of 0.1% (*w*/*v*) TCA. The homogenate was centrifuged at 12,000× *g* for 10 min at 4 °C. One mL of the supernatant was mixed with 1 mL of 20% TCA containing 0.5% (*w*/*v*) TBA (thiobarbituric acid). The mixture was heated at 90 °C for 30 min, quickly cooled in an ice bath, and then centrifuged at 12,000× *g* for 10 min. The absorbance was spectrophotometrically measured at 532 nm. The obtained absorbance was corrected by subtracting the unspecific value at 600 nm. An extinction coefficient of 155 mM^−1^ cm^−1^ was used to calculate the MDA concentration.

### 2.8. Soluble Protein and Enzyme Activity Assay

Samples (0.5 g) were ground with four volumes of a 50 mM Tris–HCl buffer, pH 8 containing 0.3 M mannitol, 1 mM EDTA, and 0.05% (*w*/*v*) cysteine. The homogenate was centrifuged at 20,000× *g* for 20 min at 4 °C. The supernatant was used for spectrophotometric analysis of the total proteins and enzymatic activities. The total protein content of samples was measured with bovine serum albumin as the standard, according to Bradford [30]. The enzymatic spectrophotometric assays for the determination of CAT (EC 1.11.1.6) and POD (EC 1.11.1.7) were performed according to the method described by Paciolla and colleagues [27] with slight modifications. Catalase activity assay was evaluated by following H_2_O_2_ dismutation at 240 nm in a reaction mixture containing 50 μg of total proteins, 0.1 M phosphate buffer, pH 7.0, and 0.88 µM H_2_O_2_. Peroxidase activity was measured using 4-methoxy-1-naphthol (4-MN) as substrate. The reaction mixture contained 50 μg of total proteins, 0.1 M Tris Acetate buffer, pH 5.0, 0.1 M 4-MN, and 10 mM H_2_O_2_ in a total volume of 1 mL. The decrease in absorbance due to the oxidation of 4-MN was measured at 593 nm.

### 2.9. Statistical Analysis

Three independent biological replicates were performed for all experiments. For the investigated analyses, means and standard deviations (±SD) are shown. The Shapiro–Wilk test was run to verify the normality of data distribution. Then, the parametric test Analysis of Variances (one-way ANOVA), followed by Tukey’s post hoc test, was used to compare the different illumination conditions using Minitab software (version 18, Minitab Inc., State College, PA, USA). Differences were considered statistically significant at a *p*-value ≤ 0.05.

## 3. Results

### 3.1. Effect of Light on Biometric Parameters

The effect of different light qualities and intensities on the root, aerial part, and total length of the seedlings are shown in Figure 1 and Table 1. Lentil seedling lengths were higher under RL treatment on both days, stimulating the growth with an average increase of 26.7% and 62% compared to D and BL. BL showed the lowest values.

Overall, the different PPFD values (100, 300, and 500 µmol m^−2^ s^−1^) of the same monochromatic LED light did not affect seedling growth. At 3 DAS, the aerial part/root ratio of seedlings grown under BL showed the highest value. This trend was the opposite at 5 DAS, where the value was higher under RL treatment. At both harvest days, the highest FW values were detected in seedlings grown under RL, compared with those grown under BL and D conditions. No significative effect of light intensity was observed for each wavelength. No significant difference in DM was observed between seedlings grown in D and BL, and the lowest values were reached in RL300 and RL500 treatments.

### 3.2. Chlorophylls and Carotenoid Content

At 3 DAS, the highest values for Chl *a*, Chl *b*, and total Chls were obtained at 100 and 500 PPFD in both wavelengths, while the lowest values were observed at 300 µmol m^−2^ s^−1^ (Table 2). No significant difference was found between BL and RL within the same intensity treatments. The content of carotenoids reached the highest value at 100 and 500 µmol m^−2^ s^−1^ in both light treatments. At 5 DAS, higher levels of Chl *a*, Chl *b*, total Chls, and Car were observed under the RL300. Interestingly, a decrease in all pigments occurred in BL with increasing light intensity. No significant differences were reported in the Chl *a*:*b* ratio between light-intensity treatments of the same light-quality treatment (Table 2).

### 3.3. Changes in Phenol Content

The effect of LED light spectra and intensity on the total phenolic content in lentil seedlings is shown in Figure 2. Lentil seedlings treated with RL500 contained less phenols than those grown in darkness and under other light conditions. The highest content of phenols was observed at 300 µmol m^−2^ s^−1^ on the second harvesting day under BL. No significant differences between BL and RL at 100 and 300 PPFD (at 3 DAS) and 100 and 500 PPFD (at 5 DAS) were detected. Overall, we observed that when growing, while the phenols increase, the dry matter content percentage decreases, with a slight negative correlation.

### 3.4. Changes in Ascorbate Pool

The changes in AsA and DHA after LED light treatments are shown in Figure 3. Lentil seedlings exposed to light showed higher AsA content compared to the darkness. A significant BL effect was detected on ascorbate reduced form, with an average increase of 35% and 50% compared to RL-grown plantlets in the two days of harvesting, respectively. The highest levels were observed at 300 and 500 µmol m^−2^ s^−1^ at both harvest times with higher values at 3 than 5 days and with no significative difference in AsA content at the two different light intensities. DHA content at 3 DAS did not significantly change among the treatments compared to the D, except in the RL300 treatment, which showed a lower value. At 5 DAS, the DHA content increased with increasing intensity levels.

The ascorbate redox ratio (AsA/AsA + DHA) was higher under light treatments than in darkness (Figure 3C,D). Generally, no significant differences were observed between BL and RL according to the same light intensity.

### 3.5. Light Influence on H_2_O_2_ and Lipid Peroxidation Levels

The H_2_O_2_ content significantly increased in the seedlings grown under the LED light treatments compared to the D (Figure 4A,B). Seedlings grown under BL had a significantly higher content of H_2_O_2_ than those grown under RL. An increase in the content of H_2_O_2_ was measured with increasing light intensity. The highest values were reached under BL500 at 3 DAS and under BL300 at 5 DAS. The levels of lipid peroxidation are shown in Figure 4C,D. At 3 DAS, under BL and RL treatments, MDA content was similar and it was observed to increase with increasing light intensity. Moreover, at 300 and 500 PPFD, MDA content was higher in the samples grown under LED lights compared to the darkness condition. Similarly, at 5 DAS, light treatments showed higher MDA content compared to the D, with RL showing a higher value than the BL treatments, at all tested light-intensity conditions.

### 3.6. Antioxidant Enzymes Activity

The activities of the antioxidant enzymes POD and CAT are shown in Figure 5. Three days after sowing, the 300 µmol m^−2^ s^−1^ treatment caused a significant decrease in the activity of both enzymes compared to the dark and other intensity treatments. At 500 µmol m^−2^ s^−1^, the activity of these two enzymes varied significantly according to the wavelength, increasing in BL-exposed seedlings. Generally, at 100 µmol m^−2^ s^−1,^ the activity reached comparable values between the two light treatments. Five days after sowing, POD activity in RL treatments significantly decreased at 300 and 500 PPFD compared to the BL treatments and its activity was comparable to the dark condition. Under BL treatments, a significant rise occurred in all tested conditions. The CAT activity was higher under BL treatment at 300 PPFD compared to the other treatments. At 500 PPFD, CAT activity was comparable to the D, and the lowest level was observed under RL treatment.

## 4. Discussion

The LED lighting system, characterized by relatively narrow-band spectra, allows the use of monochromatic light to achieve a better understanding of plant–light interaction. As optimal light conditions change among plants, such knowledge might help develop species-specific “light recipes” for agricultural production, especially in controlled environments, to optimise plant growth, development, defence mechanism, nutritional quality, and yield horticultural benefits. Therefore, in the present study, the influence of spectral composition and intensity on the development, bioactive compounds content, and cell redox state in *Lens culinaris* was investigated.

Our data underlined that light quality and intensity differently influenced the tested morphological and biochemical parameters in lentil seedlings. In particular, our experimental picture allowed us to evaluate the response of lentil seedlings to red and blue monochromatic LED light conditions in the transition from seed to seedling and early development.

The morphophysiological traits of *L. culinaris* seedlings varied in response to light quality and quantity. RL promoted epicotyl elongation, even though in the etiolated plants, generally, the elongation is more significant than in light-grown plants. In RL treatment, the dry matter content was lower compared to BL and D, and the RL-grown plantlets showed longer shoot and root, with an average increase in seedling length of 26.7% and 62% compared to D and BL, suggesting a shade avoidance syndrome-like phenotype (SAS) [31]. Low total light intensity and light quality (spectral composition) are responsible for SAS. Although the main signal to trigger shadow avoidance mechanisms is considered a low red-to-far-red light ratio, reduced blue light can also trigger SAS [32,33]. The phenotype observed under red light could be given, not so much by a direct effect of this wavelength on the elongation of the root and aerial part as by the completely missing blue light. Also, light intensities applied in this study may be perceived as limiting light conditions and trigger a SAS-like phenotype through the involvement of phytochromes, whose role deserves further investigation.

Relative to fresh weight only with RL, an increase is observed. This increase, however, was not determined by a rise in protein content, sugars, fibres, and other macronutrients but rather by the water content augmenting, as supported by the highest dry matter loss found in the samples grown under RL. Fast growth and increased water content in seedlings may indicate an influence of red light on the water uptake and activation of the metabolic system, which results in higher growth by the cell elongation and consumption of storage nutrients.

Compared to blue light, for which a promoter effect was observed, red light exerts a minor effect on enzymatic antioxidants. Relative to RL, the significant increase in MDA level at 5 DAS can be explained by the persisting H_2_O_2_ level and the lower AsA content and POD and CAT enzyme activity. This might cause lipid membrane peroxidation, which increases the cell’s oxidative status. However, no visible symptom of suffering in the shoots was observed in the seedlings grown under RL (Figure 1).

At 3 DAS, in RL, the root was longer, showing that this monochromatic light promotes the development of root apparatus. At 5 DAS, under RL treatment, the root growth rate was lower than the aerial part, as confirmed by the increase in aerial part/root ratio. Indeed, the appearance of the first leaves photosynthesizing and increasing photosynthetic pigments, photosynthetic efficiency, and sugar production balanced the growth processes of the whole plant. On the other hand, the lower aerial part/root ratio observed at 5 DAS under BL treatment suggests that this light wavelength has an inhibitory effect on plant growth.

A blue light-promoting effect was observed on most of the analyzed biochemical parameters compared to darkness and red light. BL induced a higher increase in enzymatic activities, ascorbate, and H_2_O_2_ contents. Regarding the H_2_O_2_-utilizing enzymes, CAT and POD were differently modulated by light quality and quantity. The activity of these enzymes significantly increased in BL (higher at 3 DAS), suggesting that the steady-state level of H_2_O_2_ was monitored during lighting time. It is known that H_2_O_2_ at low concentrations (plants can tolerate up to 10^2^ − 2 × 10^5^ µM) plays a key role in regulating plant growth, development, resistance responses, and signal transduction [34,35]. In addition, the higher AsA content observed in BL efficiently contributed to maintaining non-toxic levels in cells [36,37]. Indeed, AsA is utilized as a specific electron donor to reduce H_2_O_2_ into H_2_O in the reaction catalyzed by ascorbate peroxidase enzyme [38].

On the other hand, it is known that light stimulates ascorbate accumulation in plants, modulating the expression of genes involved in the ascorbate biosynthesis pathway, such as the key enzyme GDP-L-galactose phosphorylase (GGP) [39]. However, the mechanisms by which light influences ascorbic biosynthesis are still poorly understood. Recently, Bournonville et al. [40] showed that BL can affect the ascorbate biosynthesis by inactivating the PAS/LOV photoreceptor, which acts as a negative regulator of ascorbate biosynthesis by inhibiting the GGP. In this respect, the higher content of AsA in lentil seedlings grown under BL strongly suggests the positive role of BL in promoting the production of this antioxidant metabolite. The higher AsA content positively affected the cell redox state of AsA (AsA/AsA + DHA), which helps maintain the cell redox imbalance.

In the literature, it is reported that BL and high RL/BL ratio are positively correlated with carotenoid and chlorophyll content [41,42]. Besides light quality, the irradiance of light treatment appeared important for photosynthetic pigments. We observed that photosynthetic pigments content largely depended on the intensity applied, and no exclusive effect emerged between light qualities. This trend suggests that photoreceptor activation by BL and RL is differently regulated depending on the quality and quantity of light. In this regard, the use of monochromatic LED, single and matched, could contribute to identifying specific values of spectra light useful to increase the photosynthetic pigment level.

Phenolic compounds represent a broad group of phytochemicals with various biological functions that participate in response to light fluctuations in plants. The production of these secondary metabolites is affected mainly by light [11,43], as these compounds act as photoprotective and/or antioxidant compounds [44]. Swieca et al. [45] observed that phenolic compound levels in lentil seedlings grown under continuous light decrease during plant growth, reaching not different values from those obtained from etiolate seedlings. Some other studies reported that BL promoted the synthesis of phenol compounds as compared to RL [46]. Generally, as the intensity increases the phenol content increases, but this also depends on the species and stage of growth and LED exposure [47]. In the earlier stage of development, we recorded an increase compared to the dark, not reconfirmed in the older seedlings. Instead, a strong inhibitory effect of the greater light intensity (500 µmol m^−2^ s^−1^) emerged, underlining that phenols are very sensitive to the highest irradiance. At this growth stage, lentil plantlets are likely still in a stage with no adequate energy flow for this secondary metabolite pathway.

## 5. Conclusions

Light quality (spectral composition), quantity (intensity), and duration (photoperiod) vary over days, seasonal variations, and geographic locations, largely influence plant responses. Currently, the combined effect of light quality, irradiance, and photoperiod, such as single light quality effects, on plant growth, development, and physiological response are still poorly understood. Therefore, in this study, we analyzed the effect of blue and red monochromatic LED lights at the intensities of 100, 300, and 500 µmol m^−2^ s^−1^ (two monochromatic light quality × three light intensities treatments × continuous lighting) on the morphological and biochemical characteristics of lentil seedlings at the early growth stages. Blue light shows induced metabolic perturbations, which generally positively influence enzymatic (POD and CAT) and non-enzymatic (ascorbic acid and phenols) antioxidant components. Red light had a positive effect generally on the biometric parameters (length and fresh weight). Photosynthetic pigment content widely varies with the type and irradiance of light, also depending on seedling development. In light of the results obtained from this work and therefore the highlighted effect of blue light on antioxidant metabolism, and by contrast those of red light on the morphology, future investigations could concern the deepening of the action of the light conditions used on the antioxidant machinery and on the anatomical modifications, investigating the action of the photoreceptors. In this way, the results could be useful to improve the sustainability and efficiency of agricultural production and for a better understanding of the adaptation of plants to different lighting conditions.

## Figures and Tables

**Figure 1 biology-13-00012-f001:**
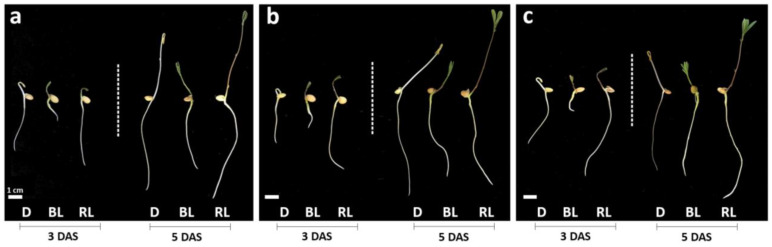
Effects of LED lights and darkness on the growth of lentil seedlings. Seedlings grown in darkness (D), blue LED light (BL), and red LED light (RL) at 3 and 5 days after sowing (DAS) with light intensities of 100 (**a**), 300 (**b**), and 500 (**c**) µmol m^−2^ s^−1^.

**Figure 2 biology-13-00012-f002:**
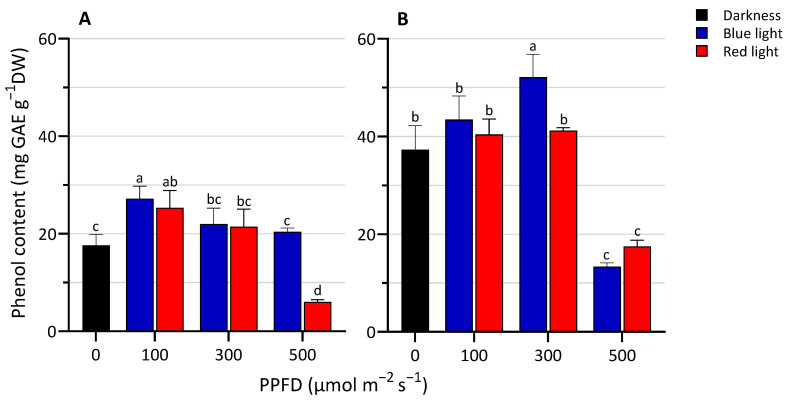
Total phenolic content at 3 (**A**) and 5 (**B**) days after sowing (DAS) of lentil seedlings grown in darkness and under blue or red LED at levels of PPFD of 100, 300, and 500 µmol m^−2^ s^−1^. The values are expressed as mean ± SD (n = 4). According to Tukey’s test, the same letters indicate non-significant differences among treatments (*p* ≤ 0.05).

**Figure 3 biology-13-00012-f003:**
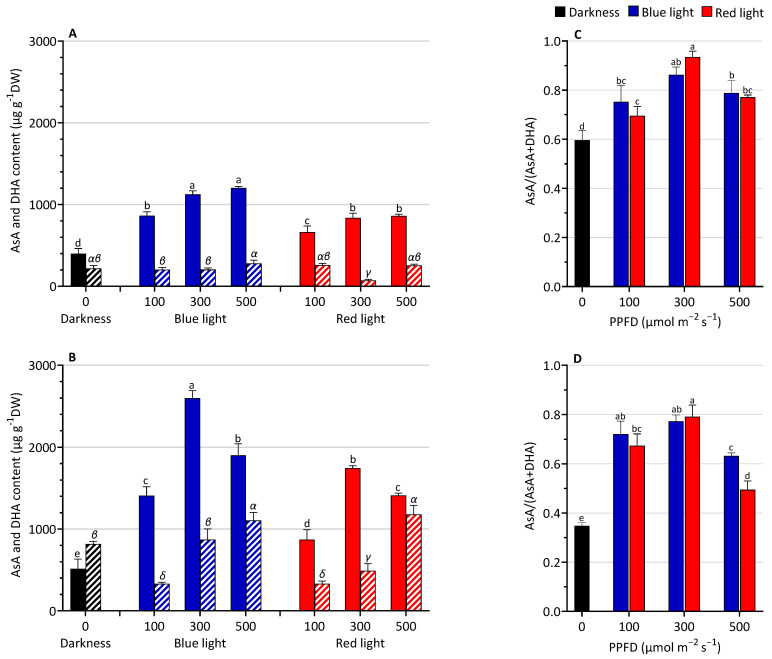
Ascorbate (AsA) and dehydroascorbate (DHA) content and AsA/AsA + DHA ratio at 3 (**A**,**C**) and 5 (**B**,**D**) days after sowing (DAS) of lentil seedlings grown under blue or red LED or in the darkness. AsA content in darkness (
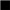
), blue LED light (
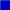
), and red LED light (
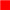
). DHA content in darkness (
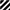
), blue LED light (
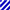
), and red LED light (
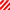
). The values are expressed as mean ± SD (n = 4). According to Tukey’s test, the same letters indicate non-significant differences among treatments (*p* ≤ 0.05). In (**A**,**B**), Latin letters are used for the statistical analysis performed on AsA content, and Greek letters refer to the one run on DHA content.

**Figure 4 biology-13-00012-f004:**
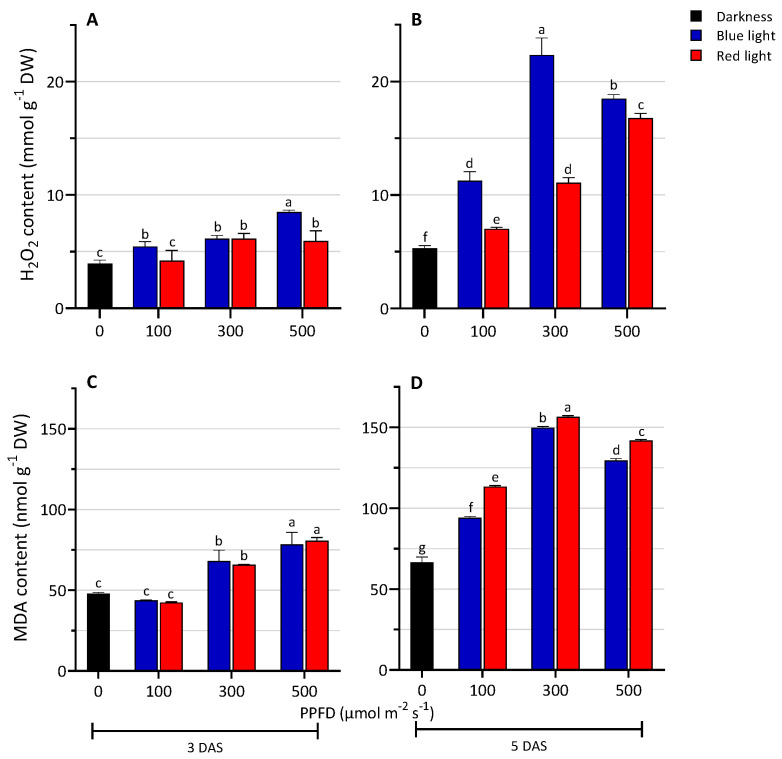
Levels of hydrogen peroxide (**A**,**B**) and lipid peroxidation (**C**,**D**) of lentil seedlings grown in darkness and under blue or red LED at levels of PPFD of 100, 300, and 500 µmol m^−2^ s^−1^ at 3 and 5 days after sowing (DAS). The values are expressed as mean ± SD (n = 4). According to Tukey’s test, the same letters indicate non-significant differences among treatments (*p* ≤ 0.05).

**Figure 5 biology-13-00012-f005:**
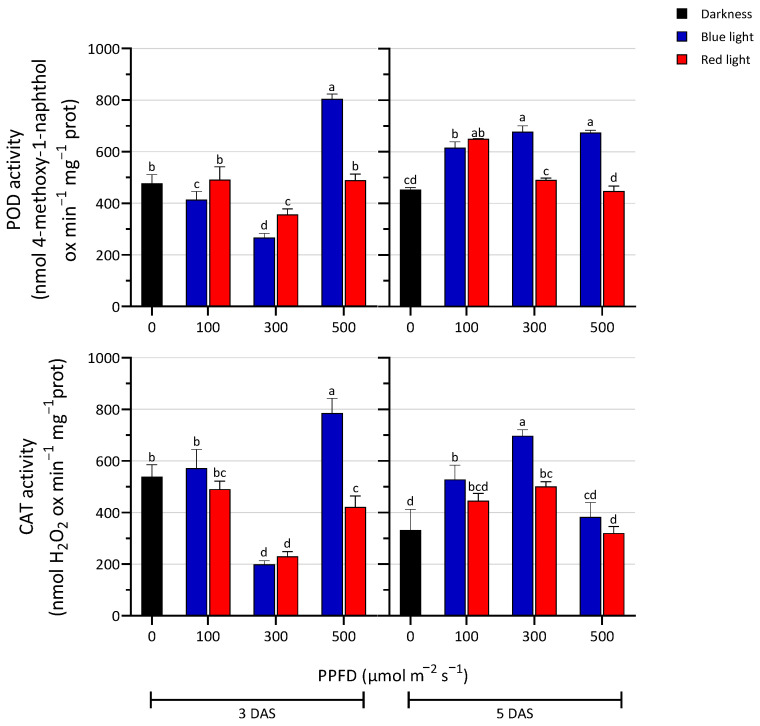
Enzyme activity of peroxidase (POD) and catalase (CAT) in seedlings treated with blue and red LED light at three PPFD levels (100, 300, and 500 µmol m^−2^ s^−1^) in the darkness, at 3 and 5 days after sowing (DAS). The values are expressed as mean ± SD (n = 4). According to Tukey’s test, the same letters indicate non-significant differences among treatments (*p* ≤ 0.05).

**Table 1 biology-13-00012-t001:** Biometric parameters of lentil seedlings at 3 and 5 days after sowing (DAS) grown under red (RL) or blue (BL) LED at three different PPFD levels (100, 300, and 500 µmol m^−2^ s^−1^) and in the darkness (D).

DAS	LED Light	PPFD (µmol m^−2^ s^−1^)	Parameters					
			Aerial Part Length (cm)	Root Length (cm)	Total Length (cm)	Aerial Part/Root Length	FW (mg)	DM (%)
	Dark	0	1.61 ± 0.38 ^b^	3.48 ± 0.77 ^c^	5.00 ± 1.17 ^b^	0.454 ± 0.081 ^cd^	100 ± 19 ^b^	26.8 ± 2.8 ^a^
3	Blue	100	1.06 ± 0.22 ^d^	1.59 ± 0.41 ^e^	2.60 ± 0.48 ^d^	0.701 ± 0.212 ^b^	93 ± 19 ^b^	25.9 ± 3.2 ^ab^
		300	1.26 ± 0.19 ^cd^	2.62 ± 0.88 ^d^	3.92 ± 1.11 ^c^	0.524 ± 0.173 ^c^	93 ± 19 ^b^	27.8 ± 2.5 ^a^
		500	1.15 ± 0.12 ^d^	1.27 ± 0.27 ^e^	2.42 ± 0.41 ^d^	0.927 ± 0.151 ^a^	92 ± 17 ^b^	26.3 ± 2.8 ^ab^
	Red	100	1.50 ± 0.18 ^bc^	4.01 ± 0.44 ^b^	5.47 ± 0.60 ^b^	0.377 ± 0.053 ^d^	97 ± 19 ^b^	23.8 ± 1.7 ^bc^
		300	1.91 ± 0.37 ^a^	5.08 ± 0.54 ^a^	7.02 ± 0.73 ^a^	0.388 ± 0.078 ^d^	104 ± 25 ^ab^	24.3 ± 2.4 ^bc^
		500	2.09 ± 0.26 ^a^	5.24 ± 0.88 ^a^	7.33 ± 1.01 ^a^	0.385 ± 0.061 ^d^	117 ± 23 ^a^	21.7 ± 1.9 ^c^
	Dark	0	3.16 ± 0.76 ^b^	6.69 ± 1.66 ^bc^	10.12 ± 2.39 ^b^	0.587 ± 0.152 ^b^	123 ± 14 ^b^	18.7 ± 2.7 ^a^
5	Blue	100	2.57 ± 0.58 ^bc^	5.95 ± 0.95 ^cd^	8.60 ± 1.30 ^c^	0.444 ± 0.081 ^c^	118 ± 19 ^b^	19.0 ± 2.2 ^a^
		300	2.03 ± 0.12 ^c^	5.93 ± 1.06 ^cd^	8.01 ± 1.12 ^c^	0.384 ± 0.069 ^c^	115 ± 17 ^b^	18.7 ± 2.0 ^a^
		500	1.94 ± 0.40 ^c^	5.28 ± 1.16 ^d^	7.26 ± 1.45 ^c^	0.415 ± 0.114 ^c^	121 ± 20 ^b^	19.0 ± 1.2 ^a^
	Red	100	5.35 ± 0.93 ^a^	7.92 ± 1.04 ^a^	13.27 ± 1.38 ^a^	0.678 ± 0.145 ^ab^	145 ± 20 ^a^	17.7 ± 1.6 ^a^
		300	5.50 ± 1.16 ^a^	7.25 ± 0.77 ^ab^	12.42 ± 1.47 ^a^	0.770 ± 0.182 ^a^	142 ± 23 ^a^	15.2 ± 2.2 ^b^
		500	4.99 ± 0.72 ^a^	7.89 ± 1.04 ^a^	12.75 ± 0.92 ^a^	0.757 ± 0.192 ^a^	145 ± 25 ^a^	13.0 ± 1.7 ^b^

Data are means ± SD (n = 6). Different letters indicate the significant difference by Tukey’s test (*p* < 0.05) on the same day under different treatment conditions (BL, RL, and D). FW: fresh weight. DM: dry matter.

**Table 2 biology-13-00012-t002:** Photosynthetic pigment content of lentil seedlings at 3 and 5 days after sowing (DAS) grown with red (RL) or blue (BL) LED at three different PPFD levels (100, 300, and 500 µmol m^−2^ s^−1^) in the darkness (D).

DAS	LED Light	PPFD (µmol m^−2^ s^−1^)	Photosynthetic Pigment Content (mg g^−1^ DW)				
			Chlorophyll *a*	Chlorophyll *b*	Total Chlorophyll	Carotenoids	Chl *a*:*b*
3	BLUE	100	82.4 ± 9.3 ^a^	33.7 ± 5.6 ^a^	116.2 ± 14.5 ^a^	47.9 ± 3.7 ^a^	2.46 ± 0.20 ^c^
		300	54.7 ± 8.2 ^b^	21.6 ± 4.1 ^c^	76.3 ± 12.3 ^b^	26.5 ± 1.2 ^c^	2.54 ± 0.11 ^bc^
		500	90.4 ± 6.0 ^a^	33.8 ± 4.5 ^a^	124.1 ± 10.4 ^a^	45.4 ± 2.9 ^a^	2.69 ± 0.17 ^abc^
	RED	100	84.1 ± 5.3 ^a^	29.2 ± 4.1 ^ab^	113.3 ± 9.4 ^a^	39.0 ± 0.2 ^b^	2.91 ± 0.24 ^a^
		300	64.1 ± 4.5 ^b^	22.3 ± 2.4 ^bc^	86.4 ± 6.3 ^b^	29.9 ± 1.9 ^c^	2.88 ± 0.27 ^ab^
		500	81.6 ± 2.4 ^a^	31.2 ± 2.8 ^a^	112.9 ± 4.9 ^a^	39.6 ± 1.3 ^b^	2.61 ± 0.17 ^abc^
5	BLUE	100	530.6 ± 11.3 ^b^	155.2 ± 16.1 ^c^	685.8 ± 26.3 ^b^	185.3 ± 8.8 ^b^	3.44 ± 0.31 ^a^
		300	439.1 ± 42.5 ^c^	133.5 ± 10.7 ^c^	572.6 ± 52.7 ^c^	160.9 ± 5.7 ^c^	3.28 ± 0.11 ^ab^
		500	323.3 ± 40.4 ^d^	102.8 ± 9.4 ^d^	426.1 ± 47.7 ^d^	131.5 ± 11.7 ^d^	3.14 ± 0.27 ^abc^
	RED	100	421.8 ± 36.2 ^c^	140.1 ± 28.4 ^c^	561.9 ± 64.4 ^c^	138.3 ± 4.7 ^d^	3.08 ± 0.43 ^abc^
		300	6316 ± 46.7 ^a^	222.9 ± 18.0 ^a^	854.5 ± 54.5 ^a^	206.7 ± 1.7 ^a^	2.84 ± 0.24 ^bc^
		500	518.0 ± 50.2 ^b^	188.6 ± 16.4 ^b^	706.6 ± 60.9 ^b^	189.9 ± 15.4 ^b^	2.76 ± 0.24 ^c^

Data are means ± SD (n = 4). Different letters indicate the significant difference by Tukey’s test (*p* < 0.05) on the same day under different treatment conditions (BL, RL, and D).

## Data Availability

Data are contained within the article.

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
