# Peer review of "Blue and Red LED Lights Differently Affect Growth Responses and Biochemical Parameters in Lentil (Lens culinaris)"

_biology, 2023, doi:10.3390/biology13010012_

Round 1
Reviewer 1 Report (Previous Reviewer 3)
Comments and Suggestions for Authors
The manuscript has improved significantly, but technical shortcomings still remain:
1. In Table 1, the indicators “Aerial part/Root length” must be rounded to thousandths (both the average and standard deviation), the indicators “FW (mg)” must be rounded to whole values (both the average and standard deviation), the indicators “DM (% )" must be rounded to the nearest tenth (both the average and the standard deviation).
2. In Table 2, the indicators “Chlorophyll a”, “Chlorophyll b”, “Total Chlorophyll” and “Carotenoids” should be rounded to the nearest tenth (both the average and standard deviation), and the indicator “Chl a:b” should be rounded to the hundredth (and average and standard deviation).
3. Under several tables, it is unclear what exactly the authors compared: “According to Tukey's test, the same letters indicate non-significant differences among treatments (p ≤ 0.05).” This comparison was carried out for example in table 1 for 14 samples located in the same column? Or does this comparison extend to 7 samples within "DAS = 3" and "DAS = 5"? This is not clear to me and will not be clear to readers. I can only guess that the second option is correct. Speculation is not allowed in the Results section of scientific articles.
4. The same applies to the note under table 2.
5. Please write in the titles of all figures and all tables in parentheses the repeatability of the studies (for example, n = 12).
Once these minor technical corrections have been made, the article may be recommended for publication.
Author Response
The manuscript has improved significantly, but technical shortcomings still remain:
- In Table 1, the indicators “Aerial part/Root length” must be rounded to thousandths (both the average and standard deviation), the indicators “FW (mg)” must be rounded to whole values (both the average and standard deviation), the indicators “DM (% )" must be rounded to the nearest tenth (both the average and the standard deviation).
We have modified all the indicators according to the reviewer’s suggestion.
- In Table 2, the indicators “Chlorophyll a”, “Chlorophyll b”, “Total Chlorophyll” and “Carotenoids” should be rounded to the nearest tenth (both the average and standard deviation), and the indicator “Chl a:b” should be rounded to the hundredth (and average and standard deviation).
We have modified all the indicators according to the reviewer’s suggestion.
- Under several tables, it is unclear what exactly the authors compared: “According to Tukey's test, the same letters indicate non-significant differences among treatments (p ≤ 0.05).” This comparison was carried out for example in table 1 for 14 samples located in the same column? Or does this comparison extend to 7 samples within "DAS = 3" and "DAS = 5"? This is not clear to me and will not be clear to readers. I can only guess that the second option is correct. Speculation is not allowed in the Results section of scientific articles.
- The same applies to the note under table 2.
We have made the procedure clearer adding a sentence in the table’s notes.
- Please write in the titles of all figures and all tables in parentheses the repeatability of the studies (for example, n = 12).
Done
Reviewer 2 Report (Previous Reviewer 2)
Comments and Suggestions for Authors
The authors have significantly corrected the article in accordance with the reviewers' comments. Almost all my recommendations have also been taken into account. In this situation, I have no objections to the publication of this manuscript. The article has been done thoroughly with a very standard set of methodological approaches. As I wrote in the first review, if the editor is satisfied with this article, it can be published and I will not object. I do not see any errors in the manuscript that would negatively affect the journal's rating. My main remark remains as it is. but I do not insist on its fulfillment
Author Response
Thank you for your comment.
Best Regards
Alessandra Villani
Reviewer 3 Report (Previous Reviewer 1)
Comments and Suggestions for Authors
The authors have done an excellent job in incorporating the necessary revisions and addressing the points raised in my review. Their revisions have significantly improved the quality and clarity of the manuscript. I am confident that accepting this paper will contribute valuable insights to the field.
Author Response
Thank you for your comment.
Best Regards
Alessandra Villani
This manuscript is a resubmission of an earlier submission. The following is a list of the peer review reports and author responses from that submission.
Round 1
Reviewer 1 Report
Comments and Suggestions for Authors
The MS entitled "Effects of blue and red monochromatic LED lighting on growth and biochemical parameters in lentil (Lens culinaris Medik.)" is an exciting theme and may interest Journal readers. The topic is original and novel. The paper closely correlates with the objective of the present research theme. The findings are well presented, the data is properly interpreted, and the conclusions are convincing. However, certain points need to be clarified before approval for publication.
1. Title: The basic concept of the paper should be clearly conveyed in the title. Consider revising it to capture the essence of the content more clearly.
2. The abstract should provide a concise overview of the paper's key points, however, currently numerical data are missing in the abstract.
2. Lines 288–296: The authors show that at different time intervals, phenolic and flavonol content increased. What is the exact effect of these on plant biomass, which is currently lacking? Consider clearly mentioning this point in the results.
3. Line 127 , at 3- and 5-days should be 3 and 5 days.
4. Line 351, Explain exact and detailed mechanism of RL on morphophysiological characteristics of lentil.
5. Consider including future research scope statement of this work.
6. The formatting of the references is inconsistent and does not follow the authors' instructions. Although the journal accepts format-free submissions, but it requires uniformity. Therefore, please use the format specified in the journal.
Author Response
REVIEWER 1
The MS entitled "Effects of blue and red monochromatic LED lighting on growth and biochemical parameters in lentil (Lens culinaris Medik.)" is an exciting theme and may interest Journal readers. The topic is original and novel. The paper closely correlates with the objective of the present research theme. The findings are well presented, the data is properly interpreted, and the conclusions are convincing. However, certain points need to be clarified before approval for publication.
- Title: The basic concept of the paper should be clearly conveyed in the title. Consider revising it to capture the essence of the content more clearly.
We agree. We modified the title based on your suggestions.
2.The abstract should provide a concise overview of the paper's key points, however, currently numerical data are missing in the abstract.
We added more details in the result of the abstract section.
- Lines 288–296: The authors show that at different time intervals, phenolic and flavonol content increased. What is the exact effect of these on plant biomass, which is currently lacking? Consider clearly mentioning this point in the results.
A sentence was added at lines 367-369
- Line 127, at 3- and 5-days should be 3 and 5 days.
Done.
- Line 351, Explain exact and detailed mechanism of RL on morphophysiological characteristics of lentil.
A deeper explanation was added at lines “336-358”.
- Consider including future research scope statement of this work.
A paragraph was added in the conclusions section at lines “429-437”
- The formatting of the references is inconsistent and does not follow the authors' instructions. Although the journal accepts format-free submissions, but it requires uniformity. Therefore, please use the format specified in the journal.
Done.
Reviewer 2 Report
Comments and Suggestions for Authors
This manuscript is devoted to the study of the effect of blue and red monochromatic LED lighting on growth and biochemical parameters in lentil (Lens culinaris Medik). Two monochromatic light quality × three light intensity treatments were tested: red light and blue light at photosynthetic photon flux density (PPFD) of 100, 300, and 500 μmol m-2 s-1. Both light quality and intensity did not affect germination. The length of seedlings growth under blue light appeared to decrease, while red light stimulated the growth. Blue light increased the content of some bioactive compounds. The authors believe that their data will be useful for optimizing the cultivation of plants under LED lighting.
The work has been done carefully. The results obtained are very nicely presented in the figures. Methodologically, the work is very simple. The authors apply long used in plant physiology methods of analysis, which give general information about the effect of LED lighting on growth and on some biochemical parameters. As it seems to me, the methods used are those that the authors are well versed in. It is not felt that the authors want to deepen their research somewhat. Light of different quality is used and the activity of photosynthetic apparatus is not characterized in any way - only the content of chlorophylls and carotenoids is determined. Using such an arrangement, it should at least say something about light receptors. It would be nice to visualize the level of transcripts that encode the enzymes being studied. It would be useful to cite literature data obtained on plants mutant for phytochromes and cyptochromes. Some results on etiolated plants would fit in very well. It would be good to at least revitalize the work in some way, using modern methods. The information obtained is new only with respect to lentils.
If the article is recommended for publication, I will note a few comments on the text of the manuscript.
-- Introduction is not written in a concentrated manner sometimes with repetitions. It would be desirable to improve this part.
-- On line 24 it says "... how light regulates plant physiology". Light cannot regulate plant physiology in any way because Plant Physiology is a branch of science.
--25-36 - "... that red vs blue light can promote the elongation of lentil sprouts..." According to the authors here it should be only red light.
--163-164 - "After 3 min, 100 mL of a 0.1 M NaOH solution..." not sure if 100 mL of 0.1 M NaOH was added.
--180 - "Lipid peroxidation was assessed by measuring the malondialdehyde (MDA) content." In fact, it is not MDA that is measured, but TBARs, secondary products of the lipid peroxidation of membranes that react with thiobarbituric acid.
--Table 1 lists Aerial part length (cm) and Root length (cm) separately and gives Total length (cm). This is actually a repetition of the results. It is the same in Table 2 for Chlorophyll a and Chlorophyll b.
If such experimental material is satisfactory to the editor, the manuscript can be published after some revision.
In my opinion, such data may be of interest for agricultural or perhaps biotechnology journals, but not for “Biology”.
Author Response
REVIEWER 2
This manuscript is devoted to the study of the effect of blue and red monochromatic LED lighting on growth and biochemical parameters in lentil (Lens culinaris Medik). Two monochromatic light quality × three light intensity treatments were tested: red light and blue light at photosynthetic photon flux density (PPFD) of 100, 300, and 500 μmol m-2 s-1. Both light quality and intensity did not affect germination. The length of seedlings growth under blue light appeared to decrease, while red light stimulated the growth. Blue light increased the content of some bioactive compounds. The authors believe that their data will be useful for optimizing the cultivation of plants under LED lighting.
The work has been done carefully. The results obtained are very nicely presented in the figures. Methodologically, the work is very simple. The authors apply long used in plant physiology methods of analysis, which give general information about the effect of LED lighting on growth and on some biochemical parameters. As it seems to me, the methods used are those that the authors are well versed in. It is not felt that the authors want to deepen their research somewhat. Light of different quality is used and the activity of photosynthetic apparatus is not characterized in any way - only the content of chlorophylls and carotenoids is determined. Using such an arrangement, it should at least say something about light receptors. It would be nice to visualize the level of transcripts that encode the enzymes being studied. It would be useful to cite literature data obtained on plants mutant for phytochromes and cyptochromes. Some results on etiolated plants would fit in very well. It would be good to at least revitalize the work in some way, using modern methods. The information obtained is new only with respect to lentils.
If such experimental material is satisfactory to the editor, the manuscript can be published after some revision.
In my opinion, such data may be of interest for agricultural or perhaps biotechnology journals, but not for “Biology”.
If the article is recommended for publication, I will note a few comments on the text of the manuscript.
- Introduction is not written in a concentrated manner sometimes with repetitions. It would be desirable to improve this part.
Introduction section was strongly modified and improved as suggested by the reviewer.
- On line 24 it says "... how light regulates plant physiology". Light cannot regulate plant physiology in any way because Plant Physiology is a branch of science.
We replaced physiology with “plant growth and development”.
- 25-36 - "... that red vs blue light can promote the elongation of lentil sprouts..." According to the authors here it should be only red light.
We made the sentence clearer (lines “40-42”).
- 163-164 - "After 3 min, 100 mL of a 0.1 M NaOH solution..." not sure if 100 mL of 0.1 M NaOH was added.
We replaced “was added” with “were added”.
- 180 - "Lipid peroxidation was assessed by measuring the malondialdehyde (MDA) content." In fact, it is not MDA that is measured, but TBARs, secondary products of the lipid peroxidation of membranes that react with thiobarbituric acid.
We changed the sentence according to the reviewer’s comment in:
“Lipid peroxidation was assessed in terms of malondialdehyde (MDA) content by measuring the TBA–MDA complex, as described by Zhang and Kirkham [27].”
- Table 1 lists Aerial part length (cm) and Root length (cm) separately and gives Total length (cm). This is actually a repetition of the results. It is the same in Table 2 for Chlorophyll a and Chlorophyll b.
We agree that it could be a bit repetitive, but we prefer to keep the three measures since the total length gives information about the whole seedling length, while in our opinion, adding root and shoot length values can make the data more exhaustive and eventually highlight differences.
Reviewer 3 Report
Comments and Suggestions for Authors
The influence of the wavelength of light and the intensity of artificial lighting on the productivity of agricultural plants is one of the priority topics for space exploration, the creation of artificial food production systems in underground conditions, etc. At the same time, this topic is relevant for improving understanding of the fundamental processes of photosynthesis and optimizing the interaction between a plant organism and its habitat. The research topics of the authors have both scientific and practical significance.
The shortcomings of the manuscript mainly concern the statistical processing of data.
1. Remove "Medik." from the title of the article. On line 28, after the first mention of the name of the plant, you need to add its Latin name with the author "Medik."
2. In table 1, I recommend adding column 2, in which to write the day of the experiment. Describing this under the table is not a very good idea.
3. The first 8 rows in Table 1 must be rounded to hundredths, not tenths (both mean and standard deviation). Accuracy in the design of tables and figures indirectly indicates accuracy during a laboratory experiment.
4. Line 237: it is not clear from the text: different letters indicate selections within one row (7 cells of the table) or two rows (14 cells of the table). It is important.
5. Notes 2-4 also apply to table 2. Here, all figures except the last column must be rounded to tenths. In the last column, the numbers must be rounded to the nearest hundredth. The absence of ± in the numbers "Chl a:b" indicates that the authors only divided the average values for the previous columns of the table. It is unacceptable. Divide "Chlorophyll a" by "Chlorophyll b" for each of the three biological replicates, and then calculate ± for "Chl a:b".
6. Statistical processing is questionable. For example, for the first line of Table 2: "90.35 ± 4.5a", "35.14 ± 3.5a" and "126.57 ± 8.2a". For "Total Chlorophyll" 126.57 is quite acceptable, but "± 8.2" is questionable. This applies to other lines as well. For example "589.72 ± 12.4a", "213.50 ± 13.5a" and "811.00 ± 39.4a". The last number "39.4" is very large. Authors need to check the original data and statistical processing for the entire article.
7. The width and height of Figure 2 should be increased by 75%. Probably the two right columns will be significantly different from each other.
8. The width and height of Figure 2 should be increased by 45%. In figure A and B, you need to put the name of the y-axis, and not just the unit of measure. The same applies to Figure 4 and Figure 5.
9. The coordinate systems in Figure 4 need to be divided into 4 separate plots due to the fact that Tukey's test results should only apply to one coordinate system. Parts of the drawing must be labeled A, B, C, D.
10. Statistical processing (correct comparison of samples) needs to be carefully checked in all figures. For example, in figure 3B, the second and seventh columns are clearly significantly different, and signed with the same letter; the fourth and ninth columns are also different, but signed the same way. There are a lot of similar obviously dubious results of statistical data processing in figures and tables.
The authors of the manuscript have performed a large amount of research, and their results will arouse interest among specialists in various biological specialties. However, the overall good impression of the article is spoiled after the discovery of technical errors in the statistical processing of data.
Author Response
REVIEWER 3
The influence of the wavelength of light and the intensity of artificial lighting on the productivity of agricultural plants is one of the priority topics for space exploration, the creation of artificial food production systems in underground conditions, etc. At the same time, this topic is relevant for improving understanding of the fundamental processes of photosynthesis and optimizing the interaction between a plant organism and its habitat. The research topics of the authors have both scientific and practical significance.
The shortcomings of the manuscript mainly concern the statistical processing of data.
The authors of the manuscript have performed a large amount of research, and their results will arouse interest among specialists in various biological specialties. However, the overall good impression of the article is spoiled after the discovery of technical errors in the statistical processing of data.
- Remove "Medik." from the title of the article. On line 28, after the first mention of the name of the plant, you need to add its Latin name with the author "Medik."
Done
- In table 1, I recommend adding column 2, in which to write the day of the experiment. Describing this under the table is not a very good idea.
We aligned table1 with table 2 adding days’ column as first one.
- The first 8 rows in Table 1 must be rounded to hundredths, not tenths (both mean and standard deviation). Accuracy in the design of tables and figures indirectly indicates accuracy during a laboratory experiment.
Done.
- Line 237: it is not clear from the text: different letters indicate selections within one row (7 cells of the table) or two rows (14 cells of the table). It is important.
In the statistical analysis we used one-way ANOVA and in Table 1, as well as in all the parameters investigated, we showed the means value with different lowercase letters which indicate significant differences at P < 0.05 between the different treatments. The revised table should be clearer because values obtained at different days are showed separately.
- Notes 2-4 also apply to table 2. Here, all figures except the last column must be rounded to tenths. In the last column, the numbers must be rounded to the nearest hundredth. The absence of ± in the numbers "Chl a:b" indicates that the authors only divided the average values for the previous columns of the table. It is unacceptable. Divide "Chlorophyll a" by "Chlorophyll b" for each of the three biological replicates, and then calculate ± for "Chl a:b".
Done
- Statistical processing is questionable. For example, for the first line of Table 2: "90.35 ± 4.5a", "35.14 ± 3.5a" and "126.57 ± 8.2a". For "Total Chlorophyll" 126.57 is quite acceptable, but "± 8.2" is questionable. This applies to other lines as well. For example "589.72 ± 12.4a", "213.50 ± 13.5a" and "811.00 ± 39.4a". The last number "39.4" is very large. Authors need to check the original data and statistical processing for the entire article.
Done
- The width and height of Figure 2 should be increased by 75%. Probably the two right columns will be significantly different from each other.
Done
- The width and height of Figure 2 should be increased by 45%. In figure A and B, you need to put the name of the y-axis, and not just the unit of measure. The same applies to Figure 4 and Figure 5.
Done
- The coordinate systems in Figure 4 need to be divided into 4 separate plots due to the fact that Tukey's test results should only apply to one coordinate system. Parts of the drawing must be labeled A, B, C, D.
Done
- Statistical processing (correct comparison of samples) needs to be carefully checked in all figures. For example, in figure 3B, the second and seventh columns are clearly significantly different, and signed with the same letter; the fourth and ninth columns are also different, but signed the same way. There are a lot of similar obviously dubious results of statistical data processing in figures and tables.
In figure 3B, the second and seventh columns are signed with the same letter but the statistics refer to the content of the two different molecules analyzed, they do not refer to the result of the same statistical analysis; in fact, we used two different character styles: normal for the AsA and cursive for the DHA. The statistical analysis of ascorbic (full-colour columns) is not linked to the statistical analysis of dehydroascorbic (pattern columns).
We added a maybe clear explanation in the figure caption and modified the figure using the Latin font for the ascorbic and the Greek font for the dehydroascorbic.